# Escape from X Chromosome Inactivation and the Female Predominance in Autoimmune Diseases

**DOI:** 10.3390/ijms22031114

**Published:** 2021-01-23

**Authors:** Ali Youness, Charles-Henry Miquel, Jean-Charles Guéry

**Affiliations:** 1Infinity-Toulouse Institute for Infectious and Inflammatory Diseases, University of Toulouse, INSERM, CNRS, UPS, 31300 Toulouse, France; Ali.youness@inserm.fr (A.Y.); charles-henry.miquel@inserm.fr (C.-H.M.); 2Arthritis R&D, 92200 Neuilly-Sur-Seine, France

**Keywords:** autoimmune diseases, sex bias, systemic lupus erythematosus, X chromosome inactivation

## Abstract

Women represent 80% of people affected by autoimmune diseases. Although, many studies have demonstrated a role for sex hormone receptor signaling, particularly estrogens, in the direct regulation of innate and adaptive components of the immune system, recent data suggest that female sex hormones are not the only cause of the female predisposition to autoimmunity. Besides sex steroid hormones, growing evidence points towards the role of X-linked genetic factors. In female mammals, one of the two X chromosomes is randomly inactivated during embryonic development, resulting in a cellular mosaicism, where about one-half of the cells in a given tissue express either the maternal X chromosome or the paternal one. X chromosome inactivation (XCI) is however not complete and 15 to 23% of genes from the inactive X chromosome (Xi) escape XCI, thereby contributing to the emergence of a female-specific heterogeneous population of cells with bi-allelic expression of some X-linked genes. Although the direct contribution of this genetic mechanism in the female susceptibility to autoimmunity still remains to be established, the cellular mosaicism resulting from XCI escape is likely to create a unique functional plasticity within female immune cells. Here, we review recent findings identifying key immune related genes that escape XCI and the relationship between gene dosage imbalance and functional responsiveness in female cells.

## 1. Introduction

In general, women have stronger immune systems than men, they also suffer more from autoimmunity and subsequent inflammation-induced tissue damage. Although the influence of environmental factors acting on genetically predisposed individuals is an important element contributing to the development of autoimmunity, emerging evidence indicates that X-linked genetic factors and sex hormone signaling may act in concert to trigger the sex-specific development of these immunological disorders [1]. Indeed, autoimmune diseases affect women and men differently. In an industrialized country, about 5% of the population suffers from autoimmune diseases, which is estimated to represent between 14 to 23 million people in the United States [2], of which women are the most affected category (70 to 80%). This is the case with scleroderma, rheumatoid arthritis (RA), multiple sclerosis (MS), systemic lupus erythematosus (SLE), Sjögren’s syndrome, which are three to seven times more common in women than men [3].

The enhanced susceptibility of women to autoimmunity correlate with the observations that women develop faster immune responses, with higher amplitude and of better quality, compared to men to various types of pathogens and vaccines [4]. Women have also higher levels of circulating antibodies, more CD4 T cells and B cells in their blood, develop more robust cytokine responses to infection, and enhanced rejection of tumors and allografts [5]. An increased immunity in females is well conserved in mammals [6]. Because of strong immune responses female experience fitness benefits of immunity, such as accelerated clearance of infection to promote survival or provision of maternal antibodies to protect offspring from lethal infection, but are also likely to suffer its costs. Although the enhanced female predominance of autoimmune diseases has come up only recently due to the increase in life expectancy and the environmental changes in developed countries associated with the decline in infectious disease prevalence [7].

The female bias in autoimmune diseases was initially mainly attributed to the female sex hormones, estrogens, yet female predominance in autoimmune diseases is frequently observed in childhood or in postmenopausal women when estrogen levels are low [3]. More recently sex chromosomes have been put forward as a genetic mechanism to explain the sex-differences in immune responses over a wide-range of age [1,8]. Understanding the genetic mechanisms by which sex chromosomes, particularly the X chromosome, contribute to influence the strength and magnitude of immunity may, therefore, have implications, not only for the treatment of these immunopathological disorders, but also for promoting optimal protective immunity in response to pathogens or for improving vaccine efficacy.

## 2. The Role of X Chromosome in Sex Bias in Autoimmune Diseases

In most eukaryotic species, the sex of the individual is determined from the heteromorphic sex chromosome pairs. The female mammals have 2 X chromosomes (homogametic) while the male is heterogametic with 1 X chromosome and 1 Y chromosome forming the pair of sex chromosomes (Figure 1). The Y chromosome is a small 59 Mb chromosome compared to the 154 Mb of the X chromosome which encodes more than 1000 genes (Figure 2). The chromosome Y contains a ‘male-determining gene’, the *SRY* gene. The presence of *Sry* in mice determines the development of the testes, and the consequent production of testosterone. In the absence of a functional *Sry* gene, the default pathway leads to the development of female-like embryos with ovaries. A transgenic mouse model, known as four core genotypes (FCG) was used to investigate the impact of sex chromosomes, regardless of the presence of male or female hormones, on the development of autoimmune disease. The deletion of Y-linked gene (*Sry*) in male mice (Y *Sry*-), prevented the development of testis resulting in mice with a female-like hormonal background similar to that of XX mice. Backcrossing these females with males expressing a *Sry* transgene on an autosome results in *Syr*+ XX or XY animals with testes. Mice carrying the XX sex chromosome complement developed increased susceptibility to lupus syndrome, compared to XY mice, regardless of the male or female hormonal environment. This model thus reveals the impact of sex chromosomes on susceptibility to the disease in a context of identical gonadal sex. In humans, the role of the X chromosome has been revealed in males with Klinefelter’s syndrome (XXY) [9,10]. These men who carry one or more extra X chromosomes have an equivalent risk to women of developing lupus or Sjögren’s syndrome [11,12]. These data suggest that the X chromosome number plays a role in the susceptibility for developing autoimmune diseases in a genetically predisposed environment. These observations led to the hypothesis that candidate genes on the inactive X chromosome could potentially contribute to the development of autoimmunity, in particular SLE and Sjögren syndrome. However, although the strength of the FCG model allows to distinguish the effects of sex chromosomes from those of sex hormones, one of its limitations is the inability to differentiate between the addition of an inactive X chromosome (Xi) or the loss of a Y chromosome.

## 3. Candidate X-Linked Genes Escaping from XCI with Possible Contribution in Autoimmune Diseases

Due to the imbalance of X chromosome between females (XX) and males (XY), regulation of X-linked gene dosage is strictly necessary. Abnormal increase in X-linked gene expression due to the presence of the extra X chromosome in females leads to developmental dysfunction [13]. Therefore, in female mammals, one of the two X chromosomes is randomly inactivated, in the early embryo, to balance the dosage of gene expression between the sexes. The process of X chromosome inactivation (XCI) results in cellular mosaicism (Figure 1), where about half of the cells in a tissue express genes from the maternal X chromosome and the other half from the paternal X chromosome [13]. However, the pseudo-autosomal regions (PAR) of the X chromosome, PAR1 and PAR2, do not undergo XCI (Figure 2). These regions carry a minority of X-linked genes with functional homologues on the Y chromosome, and these genes (e.g., *CSF2RA*, *SLC25A6* and *IL3RA* on PAR1 or *IL9R* on PAR2). Indeed, these telomeric regions are the only part sex chromosomes that can recombine in male meiosis [14]. Most of the genes outside the pseudo-autosomal regions, however, are unique to the X chromosome with two copies in women and only one in men. Most of these genes encode proteins and microRNAs known to directly or indirectly regulate the immune response [8,15,16]. Loss-of-function mutations in some of these genes (e.g., *BTK*, *WAS*, *IL2RG*, *FOXP3*) cause primary X-linked immune deficiencies [8]. The X chromosome is known to contain the largest number of immune-related genes of the whole human genome (Figure 2). The endosomal Toll-like receptor (TLR) genes, *TLR7* and *TLR8*, of the non-pseudo-autosomal region of Xp, attest to non-redundant vital functions [17]. In support of the key role of TLR7 in certain infections, a recent study has identified loss-of-function mutations of *TLR7* associated with severe forms of COVID-19 in young men, suggesting a key role of TLR7 in the protective response against SARS-CoV2 [18]. However, XCI is incomplete and 15–23% of human X-linked genes escape XCI (Figure 1), and are thus overexpressed from both the active (Xa) and inactive (Xi) X chromosome in certain tissues or individuals [19,20].

The X chromosome has recently been the subject of numerous studies aimed at understanding the role of genes on the X chromosome in the initiation and maintenance of autoimmune aggression. Therefore, it is interesting to discuss how certain X-linked genes might become overexpressed in female autoimmunity and the functional consequences associated with cell-intrinsic imbalance in X-linked gene expression. These genes are listed in Table 1.

### 3.1. The Intracellular ssRNA Sensors: TLR7 and TLR8

The involvement of Toll-like Receptors (TLR) in the recognition of self-molecules is established for the RNA-sensing receptors of innate immunity, TLR7 and TLR8, which enhanced expression is directly linked to the pathogenesis of autoimmune diseases [1,26]. Although closely located on the X-chromosome, the regulation of TLR7 and TLR8 expression is strikingly different as both receptors are not present on the same cell types and are not regulated equally by cytokines. TLR7 is expressed primarily in plasmacytoid dendritic cells (pDC), B lymphocytes and monocytes but TLR8 is preferentially expressed in monocytes, myeloid dendritic cells and neutrophils. Activation of TLR7 in monocytes preferentially promoted the expression of CD4+ T helper 17 (Th17) cell polarizing cytokines after virus infection, whereas Th1-type cytokine production and type I interferon response were dependent on TLR8 signaling [27]. Both TLR8 and TLR7 could contribute to the pathogenesis of autoimmune diseases. TLR8 is a key player in the pathogenesis of systemic Sclerosis (SSc) due to its aberrant expression in pDCs from SSc patients and its ability to exacerbate disease in mouse model of scleroderma [28]. Several studies have shown that the expression of two copies of *Tlr7* or *Tlr8* alone was sufficient to induce autoimmunity in mice [29,30,31], whereas mice deficient in *Tlr7* with a genetic background predisposing to lupus are significantly protected against autoimmunity [32]. TLR7 plays a key role in the B response in lupus, the production of anti-ribonucleoprotein (RNP) autoantibodies, as well as in the production of type I IFN (IFN-I) by pDCs [33]. We have shown that *TLR7* escapes from XCI in pDCs, B cells and monocytes from females, as well as from Klinefelter’s males (47, XXY) [23]. In addition to cells exhibiting, as expected, a mono-allelic expression of the gene carried by the X of paternal or maternal origin, we detected in all subjects, a proportion of cells exhibiting biallelic expression of *TLR7* with frequencies ranging from 7 to 45% depending on the subjects. Female B cells with escape from XCI of *TLR7* had higher TLR7 mRNA levels compared to B cells with mono-allelic expression of the gene. Remarkably, TLR7 protein expression was higher in women blood leukocytes compared to men [23], and this enhanced expression of TLR7 protein in women PBMCs was maintained over ages [34]. B cell from women stimulated through TLR7 in vitro differentiated more efficiently into CD27^hi^ plasmablasts than B cells from men, and were enriched in cells with biallelic expression of TLR7 [23]. Likewise, a positive association was also observed between *TLR7* biallelism and IgG class switching of naive B cells exposed to TLR7 agonists but not upon addition of agonists for TLR9 located on chromosome 3 [23]. Together, these results suggested a functional advantage of *TLR7* biallelic B lymphocytes at various check-point of B cell responses in response to TLR7-specific ligands [1,26].

Recently, using a single-cell RT-PCR approach to tag allelic expression of *TLR7* using SNP markers similar as the one described in [23], Hagen et al. [22] confirmed that *TLR7* escapes from XCI, leading to biallelic expression patterns in human pDCs. Biallelic expression of *TLR7* had significantly higher TLR7 mRNA transcript levels than female monoallelic-*TLR7*-expressing pDCs. IFNα and IFNβ mRNAs were more highly transcribed in female unstimulated pDCs with biallelic expression of *TLR7*, suggesting a causal link between escape of *TLR7* from XCI and higher induction of type I IFN mRNA in these pDCs [22]. Of note, in the same study bi-allelic expression of other X-linked genes (e.g., *RPS6KA3*, *CYBB*, *BTK*, *IL13RA1*) was also observed in female pDCs. Simultaneous examination of up to 3 different loci withing the same cells showed heterogenous profile of XCI escape at single-cell resolution suggested that XCI escape at multiple loci was a rare event [22].

*Tlr8*, like *Tlr7*, escapes from XCI in mice, but the functional consequences are not known [26]. In human, this raised the question whether *TLR8*, like *TLR7*, may also escape from XCI in female cells.

### 3.2. The IRF-5 Adaptor Molecule CXorf21/TASL.

The function of this gene has been recently characterized, and *CXorf21* has been renamed as “TLR adapter interacting with the SLC15A4 on the lysosome” (TASL) [25]. TASL interacts with the endolysosomal transporter SLC15A4 to activate the IRF-5 pathway following the engagement of endosomal TLR receptors (TLR7, 8 and 9). TASL shares functional homology with the adapter proteins STING, MAVS and TRIF and appears to be a central “hub” in the activation of the TLR-IRF5 pathway [25]. It has been suggested that *CXorf21*/*TASL*, which is an interferon inducible gene, has been predicted to be subject to XCI escape, although this has not been formally established yet [35]. Additionally, a genetic polymorphism in the Xp21.2 region in the exon encoding the *CXorf21* gene has been shown to be strongly associated with SLE in European populations and CXorf21 mRNA expression levels in peripheral blood cells are indicators of SLE flares [36,37]. Taken together, these observations suggest that CXorf21/*TASL* is a strong candidate gene in SLE, for which enhanced dosage in female immune cells due to XCI escape may contribute to the gender bias observed in some autoimmune diseases such as SLE.

### 3.3. Immune Cell Homing and Third Signal Delivery: CXCR3 and CD40L

CXCR3 and CD40LG are encoded by genes on the X chromosome and are both subject to XCI escape in immune cells [21]. Biallelic expression of *CXCR3* and *CD40LG* has been documented in T cells and lymphoblastoid B cells, respectively, with frequencies ranging from 4 to 5%, using single-cell RNA FISH analysis [21]. XCI escape of *CXCR3* has been elegantly confirmed in a dual reporter mouse model [38]. Female T cells with bi-allelic expression of *CXCR3* from both X chromosomes had higher CXCR3 protein levels than monoallelic cells and showed functional differences compared to mono-allelic cells, in a model of *L. mexicana* infection [38].

CD40LG is a T-cell coactivation receptor, which activate B cells and DCs through CD40 engagement, and is expressed under transcriptional control of nuclear factor of activated T-cells (NFAT). A recent study showed that a duplication of *CD40L* was associated with autoimmune manifestations, increased expression of CD40L and the functional ability of T cells to promote B cell activation and differentiation in vitro [39]. As reported for TLR7 in human B cells [23] and pDCs [22], it is likely that the escape from XCI of *CXCR3* or *CD40L* may have functional consequences for sex-specific immune functions [38,39].

### 3.4. The Histone Demethylase KDM6a (Lysine Demethylase 6A, also Known as Utx)

*KDM6a* escapes from X chromosome inactivation [24], potentially leading to higher expression of KDM6A in immune cells of females as compared with males, in human and mouse [40]. It is involved with KMT2D in the activation of gene expression by modifications of histones which makes chromatin permissive for transcription. KDM6a removes the repressive marks associated with the triple methylation of lysine 27 of histone H3 (H3K27me3) then it associates with the methyltransferase KMT2D which can add the activating marks on lysine 4 of histone H3 (H3K4me3) [40]. The role of *Kdm6a* in autoimmunity has been recently illustrated in experimental autoimmune encephalomyelitis (EAE), the mouse model of multiple sclerosis (MS). The deletion of *Kdm6a* in mouse CD4^+^ T cells diminished neuroinflammation and protected mice from MS-like symptoms in the EAE model, due to a defect in the differentiation of pathogenic T lymphocytes [41]. This observation could explain the protective effect of the diabetes drug metformin, which is known to block the demethylase activity of KDM6a. Since *Kdm6a/Utx* expression in CD4^+^ T cells promotes disease in EAE, escape from XCI and overexpression of Kdm6a in females is consistent with the increased susceptibility of women to multiple sclerosis [41]. However, *KDM6A* escape is not all bad news for women health as this gene is also a tumor suppressor. The rate of cancer in men is more than double that in women, and KDMA6A mutation are more common in men’s cancer than in women’s [42]. Scanning cancer mutations in various tumor types for genes exhibiting more mutations in men than in women led to the characterization of six genes implicated as tumor suppressors, all on the X chromosome, including KDM6A [42]. Because all these X-linked genes were reported to escape XCI, these results supported the idea that expressing spare tumor suppressors from the Xi may contribute to avoid cancer in women, as females would require two deleterious mutations to inactivate the genes. By contrast in men, with only one X, a single mutation in those genes may be sufficient to substantially contribute to the observed higher incidence of cancer in males across a variety of tumor types [42].

## 4. Role of XIST RNA Localization on the Xi in AID?

XCI is dependent on epigenetic features including the transcription of the long noncoding RNAs (lncRNA), XIST, encoded by an X-linked gene, on the future inactive X chromosome (Xi), which accumulates along this chromosome in *cis* [13]. This event initiates the silencing of X-linked genes by recruiting the polycomb repressive complex 1 (PRC1) and 2 (PRC2) which are responsible for mono-ubiquitylation of lysin 119 on the histone H2A (H2AK119ub1) and trimethylation of the lysine 27 on histone H3 (H3K27me3) on the Xi. The acquisition of these epigenetic features is involved in the stable maintenance of the inactive state [13]. Although the role of Xist in the initiation of XCI is well established; its function in XCI maintenance is debated and still unclear. Actually, earlier works have reported that global and efficient XCI can be maintained in the absence of XIST in mature differentiated cells [43,44].

Analysis of the Xi in lymphoid cells have shown that the presence of Xist RNA and H3K27me3 were present on the Xi in HSCs and CLPs, whereas these marks were absent in pre-B, and immature B cells [45]. Chromatin of the Xi progressively changes during B cell development, initiated with the loss of Xist RNA, and heterochromatin marks gradually disappear from the Xi as pro-B cells differentiate to immature B cells [45,46]. B lymphocytes lack these marks at steady state and exhibit a unique dynamic localization of these modifications to the Xi following activation. Loss of these marks on the Xi at steady state was transient, and XIST RNA relocalization together with H3K27me3 and H2AK119ub1 occurred at the Xi upon transition from quiescence to activation [21,46]. In particular, it was shown that, upon B cell activation, the XIST RNA localization to the Xi was restored in a dynamic two-step process involving the transcription factor YY1 [46]. Similar observation was made in T cells [47]. Xist RNA localization at the Xi was altered in activated T cells of SLE patients and late-stage-disease NZB/NZW F1 mice, suggesting that Xist localization to the Xi could contribute to the maintenance of dosage compensation in lymphoïdes cells [47]. However, the overall significance of Xist repositioning to the Xi in activated lymphocytes compared to naïve cells remain to be addressed.

It was hypothesized that some genes on the inactive X chromosome (Xi) are predisposed to become partially reactivated in the naïve female lymphocytes [21]. However, examination of allelic expression of putative disease-causing candidate genes like *Tlr7* failed to reveal major differences in biallelic cell frequencies between B cells from lupus-prone and wild-type mice [48]. Although the number of subjects was too low to draw any conclusions, no marked differences in *TLR7* biallelic cell frequencies was also observed in EBV-B cells between SLE sufferers and healthy women [21]. In a recent study, conditional deletion of Xist in neurons was associated with lack of expression of the repressive marks (H2AK119ub1, H3K27me3) on the Xi. Despite this, XCI was largely preserved in the vast majority of the cells, suggesting that the maintenance of transcriptional repression of the Xi is largely independent of Xist at least in neurons [49].

Whether Xist RNA relocalization to the Xi in activated T and B cells affect the expression of genes on the Xi remains to be established.

## 5. Interactions between Sex and Genetic Polymorphism: the Case of the rs179008 Polymorphism of TLR7

The effect of genotypes may be different between males and females due to certain genetic polymorphisms, or expression quantitative trait loci (eQTL), responsible for sex-specific genetic architecture [50]. The *TLR7* polymorphism rs179008 (NM_016562.3:c.32A>T) has been recently identified as a sex-specific eQTL, causing a difference in functional expression of TLR7 protein and effector function only in females [34]. The rs179008 is a SNP which introduces a substitution of a leucine for a glutamine (p.Gln11Leu) at the protein level in the leader sequence of TLR7 thereby controlling protein dosage at the translational level through the mRNA sequence itself, rather than through signal peptide function [34]. The minor allele rs179008 T impaired type I interferon production by pDCs in response to TLR7 ligands, but only in women. Thus, the sex bias in pDC function mirrored a genotype-dependent drop in TLR7 protein expression in female leukocytes, consistent with the notion that TLR7 dosage in pDCs determines type I interferon secretion [22,51]. Importantly, in a cohort of acute HIV-1 infected women, rs179008 T allele was significantly associated with lower levels of a triad of positively correlated parameters: RNA viral load, the cell-associated HIV-1 DNA reservoir, and plasma IP-10. In addition, the higher frequency of asymptomatic clinical presentations at diagnosis for the T/T homozygous genotype implies a trend for a delayed onset of the symptoms of acute infection, supporting a beneficial effect of toning down TLR7-driven IFN-I production by pDCs during acute HIV-1 infection [34]. Beside its potential effect in viral infections the rs179008 SNP has been also associated with the susceptibility to SLE [52]. By increasing the susceptibility of women to certain viral infections, such SNP could have deleterious consequences as failure to control viral infections has been associated with the onset of SLE [53,54].

## 6. Conclusions

Sex hormones play an indisputable role in the greater susceptibility of women to develop autoimmune diseases. More recently, it has been shown that in addition to sex hormones, X-linked genetic factors contribute to the functional plasticity of immune cells in females, suggesting that epigenetic mechanisms regulating the inactive X chromosome may have important implication in the enhanced female susceptibility to autoimmune diseases. However, sex chromosome and sex hormone effects cannot be regarded as totally independent. For instance, considering the nucleic acid sensor axis TLR7/9-IRF5-TASL, evidence exist suggesting a concerted action of sex hormones and X chromosome complement in TLR7-driven pDC interferogenesis [51]. One can speculate that sex hormones and sex chromosomes may independently control the protein levels of key components of the upstream and downstream signaling modules of the endosomal TLR pathway, with estrogen-signaling regulating *IRF5* mRNA expression levels whereas X-linked genetic mechanisms controlling expression of *TLR7* though XCI escape and gene dosage effects [22,23,55]. Regarding the causal relationship between XCI escape and sex-differences in the development of spontaneous autoimmune diseases, important question still remains regarding the stability of XCI escape in immune cells and the underlying molecular mechanisms controlling XCI escape at single-cell resolution. In addition, the direct causal link between escape from XCI, namely the additional expression of genes from the Xi, and the greater susceptibility to develop autoimmune diseases remains to be demonstrated and will require the implementation of new genetic tools. Lastly, examination of the epigenetic and 3D-conformation landscape of the genomic regions of candidate escape genes should shed lights on the mechanisms controlling expression of these genes from the Xi, and could open avenues for the development of novel therapeutics in autoimmune diseases associated with dosage imbalance of X-linked genes.

## Figures and Tables

**Figure 1 ijms-22-01114-f001:**
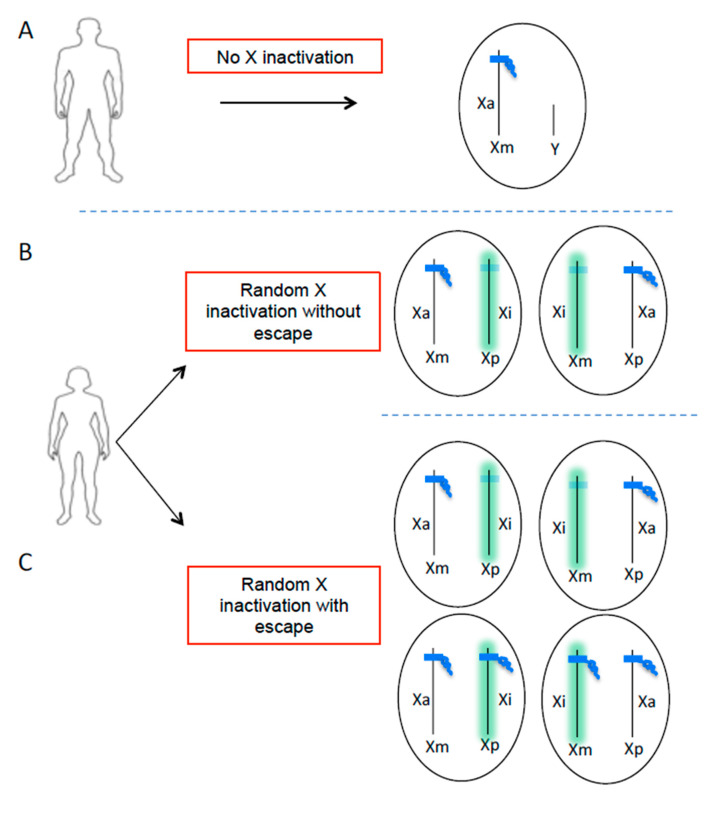
X chromosome inactivation escape generates additional cellular heterogeneity in the expression of X-linked genes in women. Men have only one active X chromosome (**A**) and women have two X chromosomes inherited from each parent (maternal X = Xm, paternal X = Xp). In order to balance the dosage of genes carried by the X chromosome between males and females, one of the two chromosomes X is randomly inactivated during embryonic development in women (**B**,**C**). This process is initiated by the long non-coding RNA XIST (green), which becomes highly expressed on one allele and coats the future inactive X chromosome (Xi) in *cis*, leading to transcriptional repression. The result is a cell mosaic where in theory half of the cells have an active chromosome (Xa) of maternal origin and the other half of paternal origin (**B**). However, certain genes can escape XCI, including *TLR7*, generating additional cellular heterogeneity in the expression of X-linked genes (**C**).

**Figure 2 ijms-22-01114-f002:**
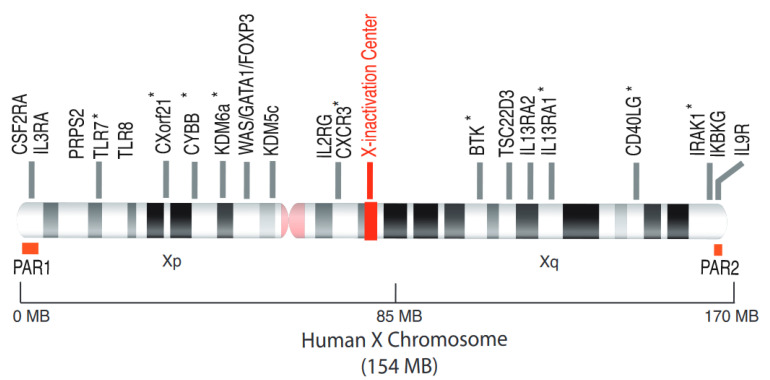
The human X chromosome and the escapee genes from XCI. The human X chromosome, 154 MB in size, codes for around 1100 genes, many of which are involved in immune responses (*WAS*, *BTK*, *FOXP3*, *IL2RG*). The *TLR7* gene, located on the short arm of the X chromosome, is able to escape from XCI. The other genes whose escape has been clearly demonstrated, are annotated with an asterisk (*) and described in the Table 1. PAR1, pseudoautosamal region 1; PAR2, pseudoautosomal region 2.

**Table 1 ijms-22-01114-t001:** Genes that escape from XCI in immune cells.

Gene Symbol	Gene Nomenclature	Cell Type	Reference
***IRAK1***	interleukin 1 receptor associated kinase 1	variable escapes in primary fibroblast cell lines	[19]
***CD40LG***	CD40 ligand	escape in activated T cells and immortalized B-cell lines generated from pediatric SLE patients or healthy females	[21]
***CXCR3***	C-X-C motif chemokine receptor 3		
***IL13RA1***	interleukin 13 receptor subunit alpha 1	escape in pDC from healthy women	[22]
***CYBB***	cytochrome b-245 beta chain		
***TLR7***	toll like receptor 7	escapes in monocyte, lymphocyte B and pDC from healthy women and Klinefelter syndrome males (XXY)	[23]
		escapes in pDC from healthy women	[22]
***KDM6a***	lysine demethylase 6A	escapes in mouse-human somatic cell hybrids	[24]
***BTK***	Burton tyrosine kinase	escapes in pDC from healthy women	[22]
***CXorf21*/** ***TASL***	chromosome X open reading frame 21; also known as TASL (TLR adaptor interacting with SLC15A4 on the Lysosome) [25]	variable escapes in primary fibroblast cell lines	[19]

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
