# Peer review of "Escape from X Chromosome Inactivation and the Female Predominance in Autoimmune Diseases"

_ijms, 2021, doi:10.3390/ijms22031114_

Round 1

Reviewer 1 Report

This a very nice work in the field.

It is very well written. However, the English used needs a fair revision. 

Perhaps, the manuscript would benefit from the add-on of some explicative figures. Please add these. 

Author Response

This a very nice work in the field.
It is very well written. However, the English used needs a fair revision. 
Perhaps, the manuscript would benefit from the add-on of some explicative figures.

We have added a new Figure (Fig. 1) to visualize the consequences of XCI escape on the cellular mosaicism and heterogeneity in women compared to men.

Reviewer 2 Report

The authors have undertaken a fairly comprehensive review of genetic factors linked to the X chromosome in sex bias in autoimmune diseases.  The authors concluded that examination of the epigenetic and 3D-conformation landscape of the genomic regions of candidate escape genes should shed lights on the mechanisms controlling expression of these genes from the inactive X chromosome, and could open avenues for the development of novel therapeutics in rheumatic diseases.  I have some comments that I believe need to be addressed prior to publication of this article.

Comments:

Page 3 line 98, “Figure 1. The Human X chromosome and the escapee genes from XCI.”, The human.

Page 4 Table 1. Please add reference numbers.

Page 5 lines 164–167, “TLR7 protein expression was higher in the leukocytes of women compared to men, and e a causal…”, Please revise this sentence.

Page 5 lines 169–171, “Recently, using a single-cell RT-PCR approach to tag allelic expression of TLR7 using SNP markers similar as the one described in (29), Hagen et al confirmed that…”, I guess (30).

Page 8 lines 317–321, “Examination of the epigenetic and 3D-conformation landscape of the genomic regions of candidate escape genes should shed lights on the mechanisms controlling expression of these genes from the Xi, and could open avenues for the development of novel therapeutics in rheumatic diseases.”, however rheumatic diseases are not discussed in the current manuscript.

Author Response

We thank the reviewer for his positive comments on our manuscript and suggestions, which have been addressed in our point-by-point responses below.

Comments:

Page 3 line 98, “Figure 1. The Human X chromosome and the escapee genes from XCI.”, The human.

It has been corrected

Page 4 Table 1. Please add reference numbers

We added the reference numbers

Page 5 lines 164–167, “TLR7 protein expression was higher in the leukocytes of women compared to men, and e a causal…”, Please revise this sentence.

The sentence has been corrected page 6 lane 191-213.

Page 5 lines 169–171, “Recently, using a single-cell RT-PCR approach to tag allelic expression of TLR7 using SNP markers similar as the one described in (29), Hagen et al confirmed that…”, I guess (30).

This has been corrected

Page 8 lines 317–321, “Examination of the epigenetic and 3D-conformation landscape of the genomic regions of candidate escape genes should shed lights on the mechanisms controlling expression of these genes from the Xi, and could open avenues for the development of novel therapeutics in rheumatic diseases.”, however rheumatic diseases are not discussed in the current manuscript.

This sentence has been modified.

Reviewer 3 Report

The manuscript entitled:" Escape from X chromosome inactivation and the female pre-dominance in autoimmune diseases" focused on the evaluation of escaping mechanism related to molecular assessment of X chromosome in autoimmune disease requires moderate revisions to be suitabel for publication.

  • In the introduction section, few details are discussed about the autoimmune mechanisms and corresponding clinical implications. In my opinion, an extensive literature revisions should be implemented in order to sole this issue. In addition, i would suggest to report some exemplificative cases of autoimmune disease and X chromosomes related genes role.
  • In the text, the authors did not exaustively explain how regulation processes (miRNA, chromatin remodelling) may be influence autoimmune disease. In my opinion, the authors should verify if these mechanisms was yet discussed in the literature.
  • In the conclusion section, the authors should also report if experimental data was identified in order to translate this contributive data in clinical practice. In particular, i would suggest if this data may play an active role in a clinical setting for specific autoimmune patients.
  • A relevant consideration may be also related to the application fo this biological model in tumor diasease where a not negligible patient's section may be eligible to Immunotherapy (IC). This revolutionary approach is based on the identification of molecular signature that allows to select patients that could benefit from drugs that active immune system against tumro cells. Please, could the authors evaluate if X chromosome escaping inactivation may be relevant in this field?

Author Response

We acknowledge the reviewer comment suggesting to add some information regarding the potential role of XCI escape in cancer. We have added the point below regarding the very interesting study by Dunford et al. Nat Genet 2016 49:10, which provided evidence that biallelic expression of X-linked tumor suppressor genes in female could explain the reduced cancer incidence in female as compared to male across a variety of tumor types.

page 8 lane 280

“However, KDM6A escape is not all bad news for women health as this gene is also a tumor suppressor. The rate of cancer in men is more than double that in women, and KDMA6A mutation are more common in men’s cancer than in women’s [42]. Scanning cancer mutations in various tumor types for genes exhibiting more mutations in men than in women led to the characterization of six genes implicated as tumor suppressors, all on the X chromosome, including KDM6A  [42]. Because all these X-linked genes were reported to escape XCI, these results supported the idea that expressing spare tumor suppressors from the Xi may contribute to avoid cancer in women, as females would require two deleterious mutations to inactivate these genes. By contrast in men, with only one X, a single mutation in those genes may be sufficient to substantially contribute to the observed higher incidence of cancer in males across a variety of tumor types [42].”